

# Effect of duration and infection control barriers of light curing unit on hardness of Bulk Fill composite resin

Xinmin He*, Denghui Zhang* and Shuli Deng

Stomatology Hospital, School of Stomatology, Zhejiang University School of Medicine, Zhejiang
Provincial Clinical Research Center for Oral Diseases, Key Laboratory of Oral Biomedical
Research of Zhejiang Province, zhejiang university, Hangzhou, Zhejiang, China
* These authors contributed equally to this work.

## ABSTRACT

**Background:** This study aimed to investigate the impact of the duration of light curing unit (LCU) usage and the use of infection control barriers on the hardness of Bulk Fill composite resin after curing. The hypotheses were that extended usage of the LCU would not reduces its output power and resin hardness, and that the presence of polyethylene film barriers exacerbates the reduction in resin hardness.

**Methods:** Based on the absence or presence of polyethylene film (PE) and the number of layers used, a 3M LED curing light (EliparTM DeepCure-S; 3M ESPE, St Paul, MN, USA) was divided into three groups: PE0, PE1, and PE3. The curing light was used 30 times daily for 20 s per exposure, at frequencies of 0, 6, and 12 months. Maximum output power tests were conducted for each group of curing lights. Custom-made plastic modules were used to stack Bulk Fill composite resin (Filtek Bulk Fill Posterior Restorative; 3M ESPE) to a thickness of 4 mm. Each group of curing lights was used to cure the modules in a direct contact manner for 20 s. Vickers hardness measurements were taken at the top and bottom surfaces of the resin specimens using a digital microhardness tester. A one-way or two-way ANOVA analyzed the power of LCUs, Vickers hardness of Bulk Fill composite resin, and hardness decrease percentage across groups. Pairwise comparisons used the Tukey test ($\alpha = 0.05$).

**Results:** As the duration of usage increased, both the power of the curing light and the hardness of the resin significantly decreased. Significant differences were observed in power and resin hardness among the PE0, PE1, and PE3 groups. When the duration of usage was 6 months or less, only multi-layered PE films led to a significant increase in the percentage decrease of hardness of cured resin from top to bottom. However, at 12 months, both single-layer and multi-layered PE films resulted in a significant increase in the percentage decrease of hardness of cured resin from top to bottom.

**Conclusion:** The output power of the light curing unit decreases with prolonged usage, thereby failing to meet the curing requirements of Bulk Fill composite resin. The use of single-layer PE as an infection control barrier is recommended.

Corresponding author
Shuli Deng, dengshuli@zju.edu.cn

## INTRODUCTION

Since the publication of the first successful case using resin-based composite materials filled by *Bowen (1963)*, light-curing composite resins have become widely utilized in clinical dentistry due to their aesthetic appearance, high plasticity, and ease of manipulation (*Fidalgo-Pereira et al., 2022*; *Guan, Zhu & Zhang, 2023*). Concurrently, light curing unit (LCU), serving as essential equipment for curing light-cured resin materials, have become indispensable in dental practice (*Lee, Young Kim & Seo, 2024*). Regular inspection and maintenance of LCUs are imperative tasks for dental care personnel (*Altaie et al., 2021*).

Traditional light-curing composite resins, when cured with LCUs, achieve a curing depth of only 2 mm per cycle. Consequently, the filling of deep cavities requires layering, resulting not only in increased chairside time but also the potential for the formation of bubbles or saliva contamination between layers (*Asyraf et al., 2023*; *El-Safty, Silikas & Watts, 2012*). To meet the demands of clinicians and the market for increased curing depth and simplified operational steps, Bulk Fill composite resin emerged. This material allows for a one-time filling of 4 mm, streamlining the procedure and saving chairside time. In addition to these benefits, Bulk Fill composite resin exhibits favorable physical and mechanical properties, including superior flexural and compressive strength, as well as excellent glossiness and wear resistance, compared to traditional composite resins (*Parra Gatica, Duran Ojeda & Wendler, 2023*). As a result, it has gained increasing favor among dentists, consequently raising the performance expectations of LCUs.

In dental treatment, to ensure the thorough curing of Bulk Fill composite resin, it is recommended to minimize the distance between the light guide tip of the LCU and the composite resin, typically within the range of 1–3 mm (*Price, Felix & Andreou, 2004*). Consequently, the light guide tip inevitably comes into contact with the patient's saliva during procedures, posing a risk of cross-contamination. LCU are not resistant to high temperatures and pressures (*Soares et al., 2020*). Given the impetus to prevent the spread of diseases such as hepatitis B, acquired immunodeficiency syndrome (AIDS), and more recently, COVID-19, the use of protective infection control barriers on such equipment has become paramount (*Dos Santos, 2021*; *Janoowalla et al., 2010*; *Verbeek et al., 2020*). However, while employing physical barriers is crucial, the impact of barrier materials on light intensity cannot be overlooked (*Khode et al., 2017*; *Soares et al., 2020*). If the output light intensity falls below the acceptable standards, it can compromise the effective curing of Bulk Fill composite resin, subsequently affecting the treatment outcomes (*Al Nahedh, Al-Senan & Alayad, 2022*). Research has shown that food wrap material can be as effective a barrier as some commercial products (*McAndrew et al., 2011*). Due to its accessibility and low cost, this study opted to utilize polyethylene film (PE), a common food wrap material, with varying layers as an infection control barrier.

This study aimed to investigate the impact of the duration of LCU usage and the use of infection control barriers on the hardness of Bulk Fill composite resin after curing. The purpose of the study was to evaluate the effects of prolonged LCU usage and the presence of PE barriers on the curing efficiency and hardness of Bulk Fill composite resin. The null

hypotheses were that: (1) The duration of LCU usage would have no effect on its power output and the hardness of the cured resin; (2) The presence of the PE as infection control barriers would have no influence on the power output or the hardness of the cured resin.

## MATERIALS AND METHODS

### Experimental grouping and LCU usage

Sixty-three new LCUs (EliparTM DeepCure-S; 3M ESPE, St Paul, MN, USA) were randomly divided into three groups based on the presence or absence of PE (Miaojie, China) and the number of film layers. Each group consisted of three LCUs. The first group, designated as the PE0 group, had no PE. The second group, named the PE1 group, had one layer of PE. The third group, denoted as the PE3 group, had three layers of PE. All LCUs underwent 30 cycles of 20-s light exposure daily to simulate the frequency of clinical usage. Subsequently, LCUs were assessed at three time points: 0, 6, and 12 months after initial usage, for further experimentation.

### LCU power testing

Maximal output power testing of all LCUs within each group was conducted using the CheckMARC® portable radiometer (BlueLight Analytics, Halifax, Canada). The strongest power mode of the LCUs was employed, with zero-distance vertical irradiation onto the CheckMARC testing device. Each LCUs underwent three repetitions. Measurements were repeated at three time points: 0, 6, and 12 months after initial usage of the light curing units.

### Resin stacking and curing

Using saliva ejectors (Medicom AMD Medicom Inc., Milan, Italy) with internal diameters of 4 mm for both top and bottom, plastic hollow cylindrical modules with a height of 4 mm were created. These modules were wrapped with PVC black electrical insulation tape (3M ESPE, Saint Paul, MN, USA) to avoid light exposure. The completed specimen modules were placed on glass slides. Bulk Fill composite resin (3M ESPE, Saint Paul, MN, USA) was filled into the modules using a resin dispenser, stacked to a thickness of 4 mm, compacted with a resin dispenser, and then lightly pressed at the top with another glass slide. The content information of Bulk Fill composite resin was provided in Table S1. Excess resin was removed using a resin dispenser. Following the instructions provided with the 3M LCU, the modules were cured using a zero-distance vertical irradiation method for 20 s (with the end face of the light guide tip in close contact with the top surface of the module). After curing, the plastic modules were cut open with a blade to remove the specimens, excess edge portions were trimmed, and the top surface was marked. Five resin specimens were stacked for each group. The cured composite resin specimens were stored in artificial saliva (D54264; Acmec biochemical, Shanghai, China) at 37 °C for 24 h, which typically contains KCl (0.4 g/L), NaCl (0.4 g/L), $CaCl_2 \cdot 2H_2O$ (0.906 g/L), $MgCl_2 \cdot 6H_2O$ (0.2 g/L), $NaH_2PO_4 \cdot 2H_2O$ (0.78 g/L), $Na_2S \cdot 9H_2O$ (0.005 g/L), Urea (1 g/L), distilled water (Balance to 1 liter) to mimic the composition of natural saliva.

## Microhardness testing of resin blocks

The resin specimens were polished through sequential grinding with silicon carbide papers (Buehler, Lake Bluff, IL, USA) of increasing fineness, followed by fine and final polishing with alumina suspension (Struers, Cleveland, OH, USA), to achieve a smooth and scratch-free surface for the Vickers micro-hardness test. Vickers hardness measurements were taken at the top and bottom of the resin specimens stacked in modules with thickness of 4 mm using the XHV-1000T-CCD Image Automatic Touch Screen Digital Microhardness Tester (Suzhou Nanguang Electronic Technology Co., Ltd, Jiangsu, China). The specimens were air-dried, and placed flat on the measurement table. A diamond cone-shaped microindenter was selected, with a test force of 500.0 g and a holding time of 15 s. Three points were selected for measurement on both the top and bottom surfaces of each specimen. Rhomboid indentations were visible on the specimen surface. Microhardness analysis was conducted using Microhardness Tester Software V3.03 (Suzhou Nanguang Electronic Technology Co., Ltd, China), employing the Vickers hardness four-point measurement method to obtain hardness values. The mean value of indentations for each sample was measured as followed equation to calculate Hv values:

$$Hv = 1{,}854.4 \times F/d^2$$

Hv is Vickers Hardness in Kg/mm2, F is load in Kg and d is diameter in mm (*Hetzner, 2003*). The average and standard deviation of Vickers hardness for the top and bottom surfaces of each resin group were calculated, along with the percentage decrease in hardness as followed equation

$$\text{the percentage decrease} = (\text{top bottom})/\text{top} \times 100\%$$

Top is Vickers Hardness of top surface of resin specimen, while bottom is Vickers Hardness of bottom surface of resin specimen. A schematic diagram of the whole research methodology is depicted in Fig. 1.

## Power analysis and statistical analysis

A power analysis using G*Power software version 3.1 (Universität Düsseldorf, Düsseldorf, Germany) determined the required total sample size to be 57 with an effect size of 0.40, $\alpha = 0.05$, and power = 0.80. Therefore, the number of resin specimens in each group was set to 21, resulting in a total sample size of 63.

A one-way ANOVA was conducted to analyze the power of the light-curing units (LCUs), the Vickers hardness of the Bulk Fill composite resin, and the percentage decrease in hardness across different groups. Pairwise comparisons were performed using the Tukey test with a significance level of $\alpha = 0.05$. Statistical analysis was performed using SPSS 20.0 (IBM, Armonk, NY, USA).

# RESULTS

## The impact of usage time on LCU output power

ANOVA showed regular power checks of the LCU revealed a significant decrease in LCU power with increasing usage duration ($p < 0.05$) (Fig. 2). The power of the new LCU was

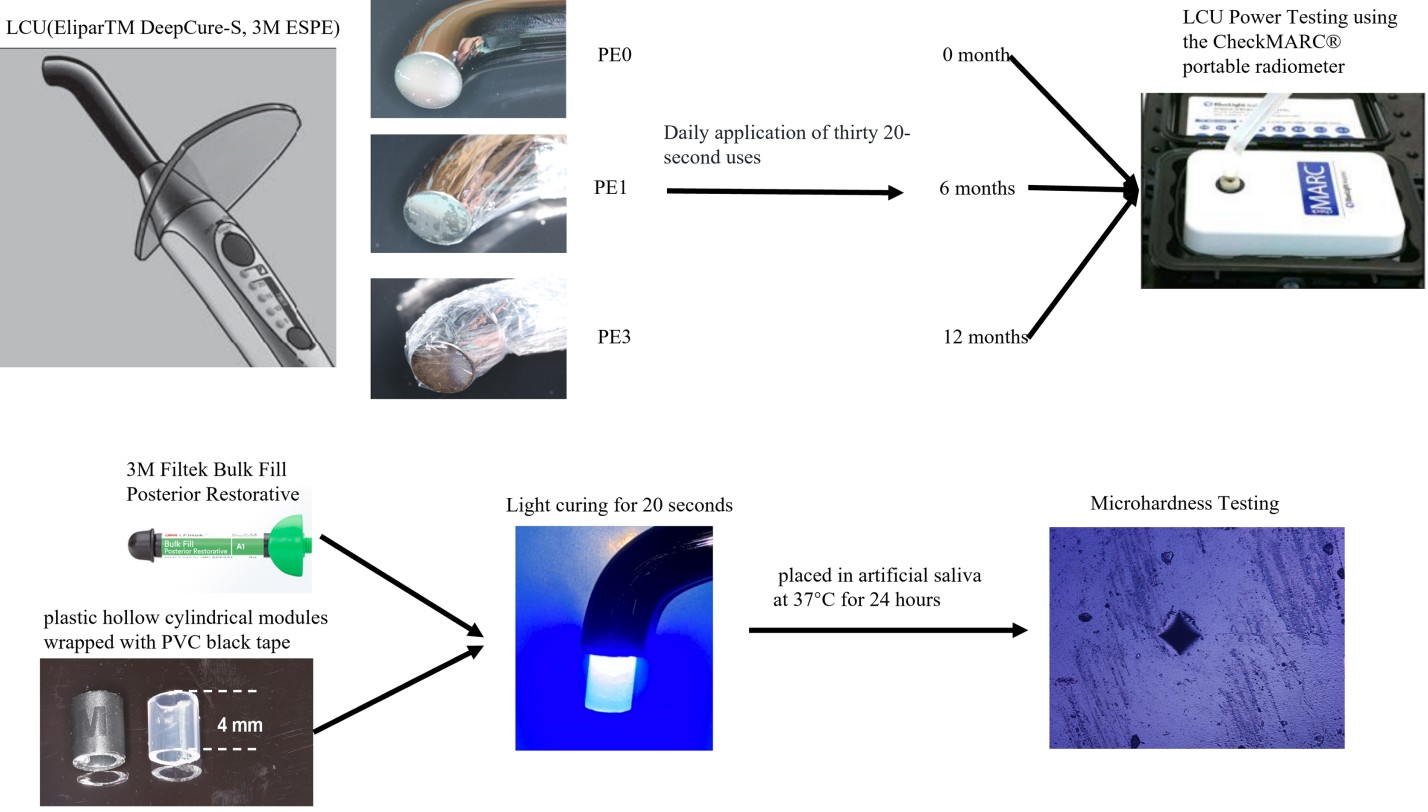

**Figure 1 Experimental grouping and procedure.** The images of the light curing unit and composite resin were photographs taken by the authors. Diagrams were sourced from the product manuals of the respective manufacturers.

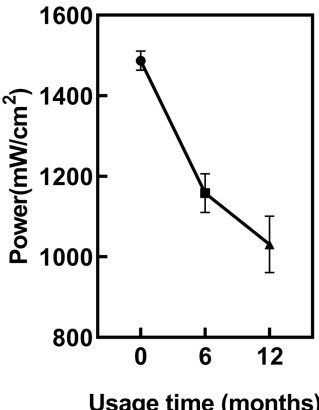

**Figure 2 The impact of usage time on LCU output power.**

1,486 ± 22 mW/cm². After 6 months of usage, the power significantly decreased to 1,158 ± 56 mW/cm² compared to new LCU ($p < 0.05$). After 12 months, it further decreased to 1,034 ± 55 mW/cm². However, the power difference between 6 and 12 months was not statistically significant ($p = 0.1159$).

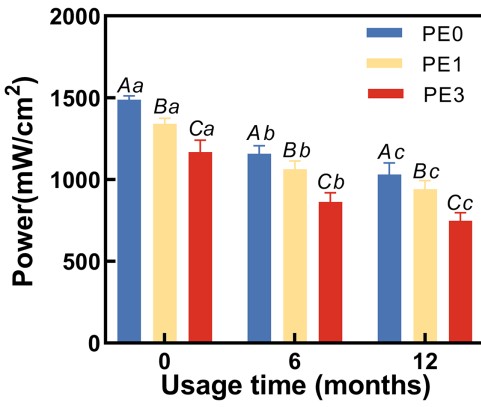

**Figure 3 The impact of infection control barrier on LCU power at different usage durations.** PE, polyethylene film. Different uppercase letters indicate a significant difference within groups of same usage time ($P < 0.05$). Different lowercase letters indicate a significant difference between groups of same PE layer ($P < 0.05$).

## The impact of infection control barrier on LCU power at different usage durations

At different usage durations, the LCU power output was tested with and without the infection control barrier PE. Two-way ANOVA showed usage time and infection control barrier both had significantly different adverse effects on the light output ($p < 0.05$). The results (Fig. 3) indicated that at 0 months of usage, the power output for PE0 was 1,486 ± 22 mW/cm$^2$, for PE1 it was 1,346 ± 43 mW/cm$^2$, and for PE3 it decreased to 1,183 ± 21 mW/cm$^2$. There were significant statistical differences among the three groups ($p < 0.05$). At 6 months of usage, the power output for PE0 was 1,158 ± 56 mW/cm$^2$, for PE1 it was 1,055 ± 44 mW/cm$^2$, and for PE3 it decreased to 861 ± 53 mW/cm$^2$. Again, significant statistical differences were observed among the three groups ($p < 0.05$). At 12 months of usage, the power output for PE0 was 1,034 ± 55 mW/cm$^2$, for PE1 it was 950 ± 57 mW/cm$^2$, and for PE3 it decreased to 754 ± 61 mW/cm$^2$. Once more, significant statistical differences were observed among the three groups ($p < 0.05$).

## The impact of infection control barrier on the hardness of Bulk Fill Resin after curing at different LCU usage durations

Two-way ANOVA showed usage time and infection control barrier had significantly different adverse effects on both top and bottom hardness of Bulk Fill Resin after curing (Figs. 4A, 4B). Figure 4A showed that at 0 months of LCU usage, the Vickers hardness at the top of the resin for the PE0 group was 114.3 ± 5.7 HV, for the PE1 group it was 104.6 ± 7.5 HV, and for the PE3 group it was 96.9 ± 4.1 HV. Only the PE3 group exhibited a significant difference in Vickers hardness at the top compared to the PE0 group ($p < 0.05$). At 6 months of LCU usage, the Vickers hardness at the top of the resin for the PE0 group was 104.2 ± 9.4 HV, for the PE1 group it was 97.3 ± 4.3 HV, and for the PE3 group it decreased to 61.4 ± 3.5 HV. The PE3 group showed significant differences in Vickers hardness at the top compared to both the PE0 and PE1 groups ($p < 0.05$). At 12 months of

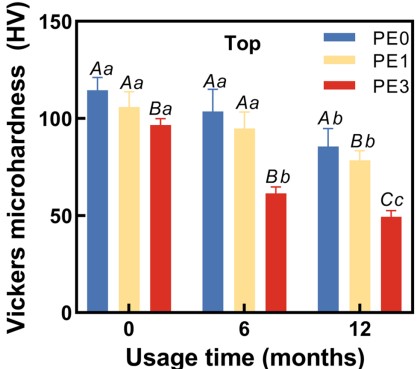
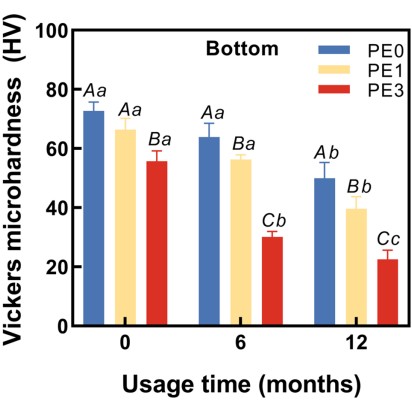
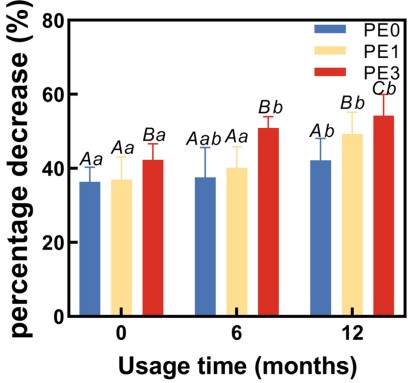

**Figure 4 The impact of infection control barrier on the hardness of Bulk Fill Resin after curing at different LCU usage durations.** (A) Vickers microhardness of top surface of cured resin. (B) Vickers microhardness of bottom surface of cured resin. (C) The percentage decrease in hardness of cured resin from top to bottom. PE, polyethylene film; Different uppercase letters indicate a significant difference within groups of same usage time ($P < 0.05$). Different lowercase letters indicate a significant difference between groups of same PE layer ($P < 0.05$).

LCU usage, the Vickers hardness at the top of the resin for the PE0 group was $86.8 \pm 9.2$ HV, for the PE1 group it was $77.2 \pm 5.4$ HV, and for the PE3 group it decreased to $49.4 \pm 2.6$ HV. Differences in Vickers hardness at the top of the resin were observed among all groups ($p < 0.05$). Comparing different usage durations (Fig. 4A), within the PE0 group, the Vickers hardness at the top of the resin cured with 12-month-old LCU showed significant differences compared to those cured with 0-month and 6-month-old LCUs ($p < 0.05$). In the PE1 group, only the Vickers hardness at the top of the resin cured with 12-month-old LCU showed significant differences compared to that cured with 0-month-old LCU ($p < 0.05$). For the PE3 group, the Vickers hardness at the top of the resin cured with LCUs of different usage durations differed significantly ($p < 0.05$).

Figure 4B showed that at 0 months of LCU usage, the Vickers hardness at the bottom of the resin for the PE0 group was $72.6 \pm 3.1$ HV, for the PE1 group it was $65.2 \pm 3.8$ HV, and for the PE3 group it was $56.4 \pm 3.5$ HV. The PE3 group exhibited significant differences in Vickers hardness at the bottom of the resin compared to both the PE0 and PE1 groups ($p < 0.05$). At 6 months of LCU usage, the Vickers hardness at the bottom of the resin for the PE0 group was $64.2 \pm 6.1$ HV, for the PE1 group it was $56.4 \pm 2.4$ HV, and for the PE3 group it decreased to $29.5 \pm 2.3$ HV. Significant differences in Vickers hardness at the bottom of the resin were observed among all groups ($p < 0.05$). At 12 months of LCU usage, the Vickers hardness at the bottom of the resin for the PE0 group was $49.7 \pm 5.4$ HV, for the PE1 group it was $40.7 \pm 2.7$ HV, and for the PE3 group it decreased to $22.1 \pm 1.8$ HV. Differences in Vickers hardness at the bottom of the resin were observed among all groups for LCUs of different usage durations ($p < 0.05$). Comparing different usage durations (Fig. 4B), within the PE0 group, the Vickers hardness at the bottom of the resin cured with 12-month-old LCU showed significant differences compared to those cured with 0-month and 6-month-old LCUs ($p < 0.05$). For the PE1 group, differences in Vickers hardness at the bottom of the resin cured with LCUs of different usage durations were observed ($p < 0.05$). For the PE3 group, the Vickers hardness at the bottom of the resin

cured with 0-month-old LCU differed significantly from those cured with 6-month and 12-month-old LCUs ($p < 0.05$).

**The impact of infection control barrier on the reduction of hardness of cured resin from top to bottom at different LCU usage durations**

The percentage decrease in hardness of cured resin from top to bottom at different LCU usage durations was calculated ((top−bottom)/top × 100%). Two-way ANOVA showed usage time and infection control barrier had significantly effects on percentage decrease in hardness of cured resin from top to bottom (Fig. 4C). The results (Fig. 4C) showed that at 0 months of usage, the hardness decrease for the PE0 group was 36.4 ± 0.4%, for the PE1 group it was 37.6 ± 3.0%, and for the PE3 group it was 41.7 ± 4.7%. There was no significant difference in the hardness decrease percentage between the PE0 and PE1 groups ($p = 0.1324$), while the hardness decrease percentage in the PE3 group differed significantly from the other two groups ($p < 0.05$). At 6 months of usage, the hardness decrease for the PE0 group was 38.4 ± 1.4%, for the PE1 group it was 42.0 ± 0.7%, and for the PE3 group it was 52.0 ± 1.1%. There was no significant difference in the hardness decrease percentage between the PE0 and PE1 groups ($p = 0.1026$), while the hardness decrease percentage in the PE3 group differed significantly from the other two groups ($p < 0.05$). At 12 months of usage, the hardness decrease for the PE0 group was 42.7 ± 1.4%, for the PE1 group it was 47.3 ± 0.9%, and for the PE3 group it was 55.2 ± 3.5%. There were significant differences in the hardness decrease percentage among all groups ($p < 0.05$). Comparing different usage durations (Fig. 4C), within the PE0 group, only the hardness decrease percentage of resin cured with the 12-month-old LCU showed a significant difference compared to that cured with the 0-month-old LCUs ($p < 0.05$). In the PE1 group, the hardness decrease percentage of resin cured with the 12-month-old LCU differed significantly from that cured with both the 0 and 6-month-old LCUs ($p < 0.05$). For the PE3 group, the hardness decrease percentage of resin cured with the 0-month-old LCU differed significantly from that cured with both the 6-month and 12-month-old LCUs ($p < 0.05$) (Fig. 4C).

## DISCUSSION

Since usage time had significantly different adverse effects on the light output (Fig. 2) and hardness of the cured resin (Fig. 4), the first hypothesis was rejected. As the infection control barrier had significantly different adverse effects on both the light output (Fig. 3) and the power output or the hardness of the cured resin, the second hypothesis was rejected (Fig. 4). In present study, we found that the output power of the light curing unit decreases with prolonged usage, thereby failing to meet the curing requirements of Bulk Fill composite resin. The use of single-layer PE as an infection control barrier is recommended.

Unlike traditional nanocomposites that require layering with each layer not exceeding 2 mm (*Besegato et al., 2019*), the Bulk Fill resin currently used in clinical practice allows for a one-time filling of 4 mm (*Silva et al., 2023*). The bulk-fill composite is selected for its high filler content and clinical performance, ensuring durability and effective curing in deep cavities (*Osiewicz et al., 2022*). Therefore, the Bulk Fill resin higher demands on the LCU.

Its widespread use and relevance in restorative dentistry make it ideal for assessing the impact of curing protocols and infection control barriers.

This study demonstrated that the LCU experience a decrease in power output over time, consequently affecting the surface hardness of Bulk Fill resin, which was line with existing studies (*Dundić et al., 2021*). Therefore, regular testing of LCU power output is warranted (*Sadeghyar, Watts & Schedle, 2020*). Should the power output fall below the clinically acceptable range, replacement of the light guide tip is necessary, or alternatively, these LCUs should be restricted to curing conventional resin rather than Bulk Fill resin. This underscores the importance for healthcare professionals to pay closer attention to the maintenance and management of light curing units (*Rohini et al., 2023*).

Contamination at the tip of the light guide of LCUs also weakens light output and diminishes resin polymerization reactions, which was also confirmed in previous study on a normal composite resin (*Hwang et al., 2012*). Therefore, in clinical practice, it is common to wrap the tip of the light guide with infection control barrier to prevent contamination (*Soares et al., 2020*). This study found that the use of infection control barriers further decreases the power output of LCUs, resulting in a noticeable decrease in the Vickers hardness of Bulk Fill composite resin after curing, with a more pronounced decrease in bottom surface hardness. Fortunately, compared to no infection control barrier, a single layer of PE had no significant effect on the surface hardness and the percentage decrease in hardness from top to bottom of resin cured by LCUs within 6 months. However, three layers of PE significantly reduced both the surface hardness of resin and the percentage decrease in hardness from top to bottom. Research also suggests that when employing an infection control barrier with low-power LCUs, caution should be exercised to avoid compromising polymerization efficiency (*Hwang et al., 2012*). Therefore, a single layer of PE can be used as an infection control barrier when the power output of the LCU is sufficient. As the power decreases, there is an increasing trend in the percentage decrease in resin hardness from bottom to top, especially when curing Bulk Fill resin using LCUs that have been in use for more than 6 months.

This reduction in curing efficacy from the top surface to the bottom surface can be attributed to the attenuation of light intensity as it penetrates deeper into the composite resin. The bulk-fill composite materials are designed to be cured in thicker layers (*Osiewicz et al., 2022*), but the light curing units may still experience a decrease in light energy as it travels through the material. This reduced light intensity at greater depths results in lower polymerization efficiency, which subsequently leads to a reduction in the hardness of the composite resin at the bottom surface compared to the top surface (*Mazão et al., 2023*). To achieve a curing depth of 4 mm for Bulk Fill resin, sufficient output power and appropriate infection control barriers are both indispensable.

This reduction in curing efficacy from the top surface to the bottom surface can be attributed to the attenuation of light intensity as it penetrates deeper into the composite resin. The bulk-fill composite materials are designed to be cured in thicker layers, but the light curing units may still experience a decrease in light energy as it travels through the material. This reduced light intensity at greater depths results in lower polymerization

efficiency, which subsequently leads to a reduction in the hardness of the composite resin at the bottom surface compared to the top surface (*Mazão et al., 2023*).

It is important to acknowledge certain limitations of this study. Firstly, the experimental setup may not fully replicate the complexities of clinical practice, as controlled laboratory conditions were utilized. Additionally, while efforts were made to mimic clinical scenarios, factors such as patient variability and intraoral conditions were not accounted for. Furthermore, the study focused solely on a specific brand and model of light curing unit, and results may vary with different equipment. Future research could explore the impact of various infection control barriers and curing conditions on different types of light curing units to provide a more comprehensive understanding of their effects on resin curing outcomes in clinical settings.

## CONCLUSIONS

The output power of the light curing unit decreases with prolonged usage, thereby failing to meet the curing requirements of Bulk Fill composite resin. The use of single-layer PE as an infection control barrier is recommended.

### Funding

This study was supported by the Key R&D program of Zhejiang (2022C03060) and General Research Project of Zhejiang Provincial Department of Education (Y201942154). The funders had no role in study design, data collection and analysis, decision to publish, or preparation of the manuscript.

### Grant Disclosures

The following grant information was disclosed by the authors:
Key R&D program of Zhejiang: 2022C03060.
General Research Project of Zhejiang Provincial Department of Education: Y201942154.

### Competing Interests

The authors declare that they have no competing interests.

### Author Contributions

- Xinmin He conceived and designed the experiments, performed the experiments, analyzed the data, prepared figures and/or tables, authored or reviewed drafts of the article, and approved the final draft.
- Denghui Zhang performed the experiments, prepared figures and/or tables, authored or reviewed drafts of the article, and approved the final draft.
- Shuli Deng conceived and designed the experiments, authored or reviewed drafts of the article, provided resources, and approved the final draft.

### Data Availability

The raw measurements are available in the Supplemental File.

## Supplemental Information

Supplemental information for this article can be found online at http://dx.doi.org/10.7717/peerj.18021#supplemental-information.

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
