# Peer review of "Effect of duration and infection control barriers of light curing unit on hardness of Bulk Fill composite resin"

_PeerJ, doi:10.7717/peerj.18021_

## Round 0.1 · original submission · Major Revisions

Dear authors,
Based on the reports of external peer reviewers, my decision is that the manuscript would benefit from a major revision. I recommend adding the objective and hypothesis of the study at the end of the introduction. Additionally, details about the preparation and composition of the artificial saliva should be provided.
Please pay attention to the reviewers' comments and questions and address them thoroughly to avoid further rounds of review.

Reviewer 1 ·

Basic reporting

In this study ‘Effect of duration and infection control barriers of light curing unit on hardness of Bulk Fill composite resin’ is evaluated. This is meaningful data for dental clinicians. I think the content needs revision.
Comments for the sections
Abstract
Statistical analysis information should be added at the end of the methods section.
Introduction
L74: The purpose of the study and the hypothesis(es) of the study should be added at the end of the introduction.

Experimental design

Materials and Methods
-Why was the hardness ratio not used in this study?
-Power analysis regarding sample size in the study should be added to the methods section.
L96: Information of a product consists of name of manufacturer, city of production, name of state if it is USA, and name of country.
L103: Have the resin composite samples been surface treated (polished, wet, etc.) after preparation?
L104: Please provide information on the content of artificial saliva.
L119: No information was given about the statistical analyzes conducted in the study. Please give information about statistical analysis at the end of the methods section.

Validity of the findings

Results
-The results of statistical analysis information should be given separately (interaction or main effects). p values ​​should be clearly stated in the findings.

Figures: -The letters in the figures are very confusing. They should be reorganised to make them easier for readers to understand. Use upper and/or lower case letters or symbols for differences within or between groups. If one group has the same letter, it makes no difference if another group has the same letter. If necessary, please provide a table (with letters). ‘a’ or ‘abcd’ or ‘abc’ are similar even if they contain the same single letter. Please change ‘Victor’ to ‘Vickers’ in the legend of Figure 4. The letters in the figures should all be revised.

Discussion

-The first paragraph of the discussion should begin with accepting or rejecting the hypothesis(es) of the study
-To demonstrate the superiority of the author's work, the results of this study in detail should be compared with similar literature of numerical analysis in dentistry.
-The features related to the selection of the bulk-fill composite should also be mentioned in the discussion.

·

Basic reporting

1. The English grammar used is quite simple and would help readers understand the manuscript. However, I would ask the author to focus on editing and formatting.

2. The introduction is well-written and explains the author's intentions. However, it would be beneficial to conclude the introduction with the aim and objective of the current study.

3. It would be helpful to rewrite the equation in Line 117 as a separate equation on a separate line.

Experimental design

Please note the following points for the study:

1. The dimensions of the samples used for the microhardness testing should be mentioned again within Section 2.4.

2. Although the author has stated that statistical analysis has been conducted, the specifics of this have not been described. It would be beneficial for the readers if the author includes this as a new section (Section 2.5).

Validity of the findings

1. The results in Section 3.1 (Lines 124 and 125) do not match the results in Section 3.2 (Lines 132 and 135). Please correct them to align with the results in Section 3.2 for consistency.

2. In the discussion section, please add an explanation for why the power is reducing in this case. Listing out factors would be beneficial for dental clinicians and researchers.

3. Please merge the paragraph after Line 225 with the above paragraph to improve continuity for the readers in understanding the reduction in surface hardness.

4. Additionally, please add a paragraph explaining why there is a reduction in curing when comparing the top surface to the bottom surface.

5. Figure 3 and Figure 4 are confusing in terms of the alphabetical categorization in the statistical study. This should be explained better to help the readers understand the statistical similarities.

Additional comments

Overall, the work is well written. Please make the necessary changes to improve the article.

---

## Round 0.2 · Minor Revisions

Dear authors

One of the peer reviewers has provided some feedback, raising a few minor comments and suggestions. It would be helpful if you could take the time to review and address these comments to ensure the final version meets all the reviewers' expectations.

Reviewer 1 ·

Basic reporting

All concerns have been revised by the authors compared to the previous version. But some minor edits should be revise.
-Please change ‘Victor’ to ‘Vickers’ in the legend of Figure 4.
-Please change "two way" to "two-way".
-Specify in a separate table the content information of the resin composite (Filtek Bulk Fill) used in the study.

Experimental design

All concerns have been revised by the authors compared to the previous version.

Validity of the findings

All concerns have been revised by the authors compared to the previous version.

·

Basic reporting

The author has made the necessary revisions to the manuscript.

Experimental design

The author has made the necessary revisions to the manuscript.

Validity of the findings

The author has made the necessary revisions to the manuscript.

---

## Round 0.3 · accepted · Accept

Dear Authors,

After careful consideration of your revised manuscript, we are pleased to inform you that your submission has been accepted for publication.

We appreciate the effort you have put into addressing the feedback, and we believe your work will make a valuable contribution.

Thank you for your cooperation and patience throughout the review process.

Sincerely,

Reviewer 1 ·

Basic reporting

All concerns have been revised by the authors compared to the previous version. Thank you.

Experimental design

All concerns have been revised by the authors compared to the previous version. Thank you.

Validity of the findings

All concerns have been revised by the authors compared to the previous version. Thank you.

Additional comments

All concerns have been revised by the authors compared to the previous version. Thank you.